



**High number concentrations of transparent exopolymer particles (TEP) in**
**ambient aerosol particles and cloud water – A case study at the tropical**
**Atlantic Ocean**
**Manuela van Pinxteren[1], Tiera-Brandy Robinson[2], Sebastian Zeppenfeld[1], Xianda Gong[3+],**
**Enno Bahlmann[4], Khanneh Wadinga Fomba[1], Nadja Triesch[1], Frank Stratmann[3], Oliver Wurl[2],**
**Anja Engel[5], Heike Wex[3], Hartmut Herrmann[1]***
*Corresponding author: Hartmut Herrmann (herrmann@tropos.de)
[1] Atmospheric Chemistry Department (ACD), Leibniz-Institute for Tropospheric Research
(TROPOS), 04318 Leipzig, Germany
[2] Institute for Chemistry and Biology of the Marine Environment, Carl-von-Ossietzky
University Oldenburg, 26382 Wilhelmshaven, Germany
[3] Dept. of Experimental Cloud and Microphysics, Leibniz-Institute for Tropospheric Research
(TROPOS), 04318 Leipzig, Germany
+ now at: Center for Aerosol Science and Engineering, Department of Energy, Environmental
and Chemical Engineering, Washington University in St. Louis, 63130, MO, USA
[4] Leibniz Centre for Tropical Marine Research (ZMT), 28359 Bremen, Germany
[5] GEOMAR Helmholtz Centre for Ocean Research, Kiel 24105, Germany





Abstract
Transparent exopolymer particles (TEP) exhibit the properties of gels and are ubiquitously
found in the world oceans. Possibly, TEP may enter the atmosphere as part of sea spray
aerosol. Here, we report number concentrations of TEP (diameter > 4.5 µm) in ambient aerosol
and cloud water samples from the tropical Atlantic Ocean as well as in generated aerosol
particles using a plunging waterfall tank that was filled with the ambient sea water. The
ambient TEP concentrations ranged between $7 \times 10^2$ and $3 \times 10^4$ #TEP m$^{-3}$ in supermicron
aerosol particles and correlations to sodium ($Na^+$) and calcium ($Ca^{2+}$) ($R^2 = 0.5$) suggested some
contribution via bubble bursting. Cloud water TEP concentrations were between $4 \times 10^6$ and
$9 \times 10^6$ #TEP L$^{-1}$ corresponding to equivalent air concentrations of $2 - 4 \times 10^3$ #TEP m$^{-3}$. The TEP
concentrations in the tank-generated aerosol particles, produced from the same waters and
sampled with an equivalent system, were significantly lower ($4 \times 10^2 - 2 \times 10^3$ #TEP m$^{-3}$)
compared to the ambient concentrations.
Based on $Na^+$ concentrations in seawater and in the atmosphere, the enrichment factor for
TEP in the atmosphere was calculated. The tank-generated TEP were enriched by a factor of
50 compared to sea water and, therefore, in-line with published enrichment factors for
supermicron organic matter in general and TEP specifically. TEP enrichment in the ambient
atmosphere was on average $1 \times 10^3$ in cloud water and $9 \times 10^3$ in ambient aerosol particles and
therefore about two orders of magnitude higher than the corresponding enrichment from the
tank study. Such high enrichment of supermicron particulate organic constituents in the
atmosphere is uncommon and we propose that atmospheric TEP concentrations resulted
from a combination of enrichment during bubble bursting transfer from the ocean and TEP in-
situ formation in atmospheric phases. Abiotic in-situ formation might have occurred from
aqueous reactions of dissolved organic precursors that were present in particle and cloud
water samples, while biotic formation involves bacteria, which were abundant in the cloud
water samples.
The ambient TEP number concentrations were two orders of magnitude higher than recently
reported ice nucleating particle (INP) concentrations measured at the same location. As TEP
likely possess good properties to act as INP, in future experiments it is worth studying if a
certain part of TEP contributes a fraction of the biogenic INP population.

Keywords: Transparent exopolymer particles, marine aerosol particles, cloud water, plunging
waterfall tank, ice nucleating particles, Atlantic Ocean, Cape Verde Atmospheric Observatory
(CVAO)




## 1 Introduction

In marine ecosystems, polymer gels and gel-like material play an important role in the biochemical cycling of organic matter (OM) (Passow, 2000, 2002b). One type of gel-like particles, transparent exopolymer particles (TEP), have increasingly received attention. TEP exist as individual particles rather than diffuse exopolymeric organic material and are operationally defined as particles that are stained on 0.2 or 0.4 µm pore-sized polycarbonate filters with the dye Alcian Blue (Passow, 2002b). TEP have shown surface-active properties and are highly hydrated molecules (Passow et al. 2002a). Chemically, they consist of polysaccharide chains including uronic acids or sulphated monosaccharides that are bridged with divalent cations (mostly calcium) (Alldredge et al., 1993;Bittar et al., 2018).

In contrast to solid particles, TEP contain properties of gels; with similar constituents (carrageenans, alginic acid, and xanthan) to those that form gels, spontaneously forming from dissolved fibrillar colloids, and they can be broken up by Calcium chelators such as EDTA. However, because TEP have not yet been seen to undergo phase transition they can officially only be classified as gel-like particles (Verdugo et al., 2004). Regardless though, TEP have been shown to be highly important in sedimentation processes and carbon cycling in the sea (Mari et al., 2017), as well as highly prevalent in the sea surface microlayer (SML) (Robinson et al., 2019a) with a potentially significant effect on air-sea release of marine aerosols.

Generally, TEP can be formed via two pathways. First, the biotic pathway happens via a breakdown and secretion of precursor material from an organism or via a direct release as particles from aquatic organisms, e.g. as metabolic-excess waste products when nutrients are limited (Decho and Gutierrez, 2017;Engel et al., 2004;Engel et al., 2002). High TEP concentrations are usually associated with phytoplankton blooms, with the majority of precursor material being released by diatoms and to a lesser extent other plankton species. However, bacteria are also associated with TEP production, although their exact role is still not resolved (Passow, 2002a). Secondly, TEP form through abiotic pathways. These could be spontaneous formation from dissolved organic precursors (e.g. dissolved polysaccharides) that are released by aquatic organisms. The abiotic formation is enhanced by turbulent or laminar shear (Engel et al., 2002;Passow, 2000). Recent studies confirmed that higher wind speeds, forming breaking waves, could be an effective transport and formation mechanism for TEP to the ocean surface (Robinson et al., 2019b).

TEP are highly sticky and provide surfaces for other molecules and bacterial colonization (Passow, 2002b), with between 0.5 and 25% (on average 3%) of marine bacteria being attached to TEP (Busch et al., 2017). TEP naturally aggregate to other particles or highly dense matter and can sink in the ocean to contribute to downward carbon fluxes. However, TEP which are not attached to sufficiently dense material will have a resulting low density and rise to the surface to form or stabilize the SML which links the oceans with the atmosphere (Wurl and Holmes, 2008).

From the ocean surface, TEP have the potential to be transferred to the atmosphere. Recently, high TEP mass concentrations of 1.4 µg m$^{-3}$ were reported in ambient marine aerosol



particles measured in a size range between 0.1 and 1 µm, suggesting that gel-like particles can
constitute more than half of the particulate OM mass (Aller et al., 2017).

Ocean-derived OM, of which TEP is a part, has been reported to be enriched and
selectively transferred (compared to sea salt) to the atmosphere (Facchini et al., 2008;Keene
et al., 2007;van Pinxteren et al., 2017). Compared to seawater concentrations, organic mass
in submicron aerosol particles is strongly enriched by factors of $10^3$ and $10^4$ (partly up to $10^5$)
(Quinn et al., 2015 and references therein) due to (not yet in detail resolved) processes during
the rise and burst of bubbles at the ocean surface (Blanchard, 1975). The enrichment of OM
in supermicron aerosol particles is significantly lower, with average aerosol enrichment factors
of $10^2$ (Hoffman and Duce, 1976;Keene et al., 2007;Quinn et al., 2015). Aerosol enrichments
have been studied for several organic compound groups such as lipids, carbohydrates, and
proteins (e.g. Gao et al., 2012;Rastelli et al., 2017;Schmitt-Kopplin et al., 2012;Triesch et al.,
2021a;Triesch et al., 2021b;Zeppenfeld et al., 2021). However, at current, data for TEP
enrichment in the atmosphere are scarce. Aller et al. (2017) presented TEP mass
concentrations in size-resolved aerosol particles and found them to contain more TEP for
submicron sizes than for larger sizes. Kuznetsova et al. (2005) reported TEP enrichment of a
factor of 40 in freshly produced sea spray. Besides TEP, other types of gel-like airborne
particles in the size range of 100 – 300 nm (and even smaller) have been observed, e.g. in the
Arctic atmosphere likely originating from the ocean surface (Bigg and Leck, 2008;Leck and
Bigg, 2005b, a).

In addition to an oceanic transfer, atmospheric in-situ formation might contribute to
OM abundance in the atmosphere. Ervens and Amato (2020) provided a framework to
estimate the production of secondary biological aerosol mass in clouds by microbial cell
growth and multiplication. It was recently shown that this pathway might represent a
significant source of biological aerosol material (Ervens and Amato, 2020;Khaled et al.,
2021;Zhang et al., 2021). In another recent study, cloud water in-situ formation of amino acids
resulting from biotic and abiotic processes has been measured and modelled (Jaber et al.,
2021). Moreover, a higher microbial enzymatic activity on the aerosol particles compared to
seawater was observed and it was hypothesised that after ejection from the ocean, active
enzymes can dynamically influence the OM concentration and composition of marine aerosol
particles (Malfatti et al., 2019). Still, the atmospheric in-situ formation of important OM
compounds and its importance is not well investigated to date and no studies exist about
atmospheric in-situ TEP formation.

Regarding the properties of ocean-derived OM in the atmosphere, its ability to act as
cloud condensation nuclei (CCN) (Orellana et al., 2011;Sellegri et al., 2021) or ice nucleating
particle (INP) (Burrows et al., 2013;Gong et al., 2020a;McCluskey et al., 2018a;McCluskey et
al., 2018b) is not well understood at present. Bigg and Leck, (2008) and Leck and Bigg (2005a)
demonstrated, based on morphology and chemical properties, that the biogenic particles
collected in air and in the surface microlayer could be consistent with polymer gels. For regions
that generally show a low total particle number concentration and low CCN (such as the high


Arctic), it was suggested that microgels are CCN (Leck and Bigg, 2005b, a;Orellana et al., 2011),
due to their hydrated and hygroscopic nature and due to the absence of other significant
aerosol particle sources.
In addition, oceanic biogenic INP sources have been discussed (Creamean et al.,
2019;Hartmann et al., 2020;Wilson et al., 2015;Zeppenfeld et al., 2019). In regions, however,
where other sources dominate, oceanic sources might not suffice to explain the INP
population, and non-marine sources most likely significantly contributed to the local INP
concentration (Gong et al., 2020a). According to their structure, biopolymers consisting of
proteins, lipids, and higher saccharides have been shown to play a role in the ice-nucleating
activity (Pummer et al., 2015). In this context, TEP might provide excellent functionalities to
act as INP, as they form a 3D network where water molecules can attach, providing a
structured surface for ice formation. A direct link between TEP and INP, however, has not yet
been experimentally shown in field studies.
Within the present study, the number concentrations and size distributions of TEP in
the ambient atmosphere in the tropical Atlantic Ocean were elucidated. We aimed at
investigating the TEP number concentrations in the ambient aerosol particles and cloud water
and to derive connections to oceanic transfer and potential in-situ formation mechanisms.
Finally, we compared the TEP number concentrations with recently published atmospheric
INP number concentrations at the same location (Gong et al., 2020a) and analyse possible
interconnections. To our knowledge, this is the first study with detailed measurements of TEP
number size distribution in different atmospheric marine compartments in the tropical
Atlantic environment.

2 Material and methods
2.1 Measurement site and ambient sampling
Samples were taken during the MarParCloud: "Marine biological production, organic
aerosol particles and marine clouds: a Process chain" campaign that took place from
September 13th to October 13th 2017 at the Cape Verde archipelago Island Sao Vicente located
in the Eastern Tropical North Atlantic (ETNA). A detailed overview of the campaign,
background, goals, and first results is available in van Pinxteren et al. (2020). Measurements
were performed at the Cape Verde Atmospheric Observatory (CVAO) as described in more
detail elsewhere (Triesch et al., 2021a;Triesch et al., 2021b;van Pinxteren et al., 2020). The
CVAO is located directly at the shoreline at the northeastern tip of the São Vicente island at
10 m a.s.l (Carpenter et al., 2010;Fomba et al., 2014). Due to the trade winds, this site is free
from local island pollution and provides reference conditions for studies of ocean-atmosphere
interactions as there is a constant north-westerly wind from the open ocean towards the
observatory. However, it also lies within the Saharan dust outflow corridor, and mainly in the
winter months (January and February), dust outbreaks frequently occur.





Total suspended aerosol particle (TSP) for TEP analysis and $PM_{10}$ sampling for analysis

of further aerosol constituents (inorganic ions, INP, dust) was performed on top of a 30 m

sampling tower of the CVAO. Tower measurements there mainly represent the conditions

above the ocean because the internal boundary layer (IBL), which can form when air passes a

surface with changing roughness (i.e. the transfer from open water to island), is mainly

beneath 30 m (Niedermeier et al., 2014). During the MarParCloud campaign, the marine

boundary layer (MBL) was well mixed as indicated by an almost uniform particle number size

distribution within the MBL (Gong et al., 2020b;van Pinxteren et al., 2020). Information on the

meteorological conditions during the sampling period is given in **Tab. S1**.

TSP were sampled with a filter sampler consisting of a filter holder equipped with a

0.2 µm pore-sized, acid-cleaned polycarbonate (PC) filter mounted to a pump. Sampling

usually took place for 24 h and the flow of the pump was between 5 and 10 L min$^{-1}$ and

frequently measured with a flowmeter. Total volumes between 10 and 15 m$^3$ were sampled.

In seawater TEP analysis, filtration is usually performed at a gentle pressure of 0.2 bar (Engel,

2009) which corresponds to a max flow rate of 21 or 38 L min$^{-1}$. The flow rate of aerosol

sampling was max. 10 L min$^{-1}$ and therefore TEP losses during aerosol particle sampling were

not expected.

$PM_{10}$ particles were sampled with a high volume sampler (Digitel, Riemer, Germany)

equipped with preheated  (105 °C for 24 h) 150 mm quartz fiber filters (Munktell, MK 360) at

a flow rate of 700 L min$^{-1}$, described in detail elsewhere (van Pinxteren et al., 2020). The

sampling times for TSP as well as $PM_{10}$ were usually set to 24 h.

Cloud water was sampled on Mt. Verde, which is the highest point of the São Vicente

Island (744 m), situated in the northeast of the Island (16°52.11ʹN, 24°56.02ʹW) and northwest

to the CVAO (van Pinxteren et al., 2020). Again, Mt. Verde experiences direct trade winds from

the ocean with no significant influence of anthropogenic activities from the island (Carpenter

et al., 2010). Bulk cloud water was collected using a compact Caltech Active Strand Cloudwater

Collectors (CASCC2) equipped with acid cleaned Teflon®strands (508 µm diameter). Cloud

droplets were caught on the strands and gravitationally channelled into an acid-precleared

Nalgene bottle. The 50% lower size cut for the CASCC2 is approximately 3.5 µm diameter.

Much of the liquid water content (LWC) in clouds is contained of drops between 10 and 30 µm

diameter and the CASCC2 is predicted to collect drops in this size range with an efficiency

greater than 80% (Demoz et al., 1996).

Three cloud water samples collected on the 20.09.2017, the 28.09.2017, and the

4.10.2017 were analysed for the TEP number concentrations. They were filtered (150-200 mL)

through 0.2 µm pore-sized, acid-cleaned filters for TEP analysis using the same filter type and

conditions as applied for the aerosol particle staining.


2.2. Particle sampling from the plunging waterfall tank



To investigate a direct oceanic transfer of TEP via bubble bursting, TSP particles were
sampled from a plunging waterfall tank experiment that is described in detail in the
MarParCloud overview paper (van Pinxteren et al., 2020, SI section). The tank was designed
to study the bubble-driven transfer of organic matter from the bulk water into the aerosol
phase. It consists of a 1400 L basin with a 500 L aerosol chamber on top. The bubble driven
transport of organic matter was induced using a skimmer on a plunging waterfall. A stainless
steel inlet was inserted in the headspace of the tank and connected with three filter holders
for offline aerosol particle sampling without size segregation (TSP). The filter system for TEP
analysis was equipped with a 0.2 µm pore-sized, acid-cleaned polycarbonate (PC) filter
mounted to a pump. Sampling usually took place for ~ 24 h, the flow of the pump was between
5 and 10 L min$^{-1}$ and frequently measured with a flowmeter. Total volumes between 9 and
10 m$^3$ were sampled. The sampling procedure was therefore identical to the ambient TEP filter
sampling. Another filter holder was equipped with a preheated 47 mm quartz fiber filter
(Munktell, MK 360) for sodium analysis. The stainless steel inlet was additionally connected
to a TROPOS-type Scanning Mobility Particle Sizer (Wiedensohler et al., 2012) for online
aerosol measurements. This method of aerosol generation resulted in an efficient generation
of nascent sea-spray aerosol particles with an aerosol particle size distribution centred around
100 nm (van Pinxteren et al. 2020).

2.3 Analysis
The filters obtained from ambient and tank-generated TSP aerosol particle sampling
and cloud water filtrations were stained with 3 mL of an Alcian blue stock solution stained
(0.02 g Alcian blue in 100 mL of acetic acid solution, pH 2.5) for 5 s yielding an insoluble non-
ionic pigment and afterward rinsed with milliQ water. The dye Alcian blue consists of a
macromolecule with a central copper phthalocyanine ring linked to four isothiouronium
groups via thiolether bonds (Passow and Alldredge, 1995). The isothiouronium groups are
strong bases and account for the cationic nature. The exact staining mechanism is not resolved
but it is believed that the cationic isothiouronium groups bond via electrostatic linkages (ionic
bonds) with the polyanionic molecules of the TEP molecule, hence the carboxylic and sulfonic
side groups are stained. Alcian Blue can also react with carbohydrate-conjuncted proteins at
proteoglycans, but not with nucleic acids and neutral biopolymers (Villacorte et al., 2015).
After staining the filters were kept at -20°C and transported to the laboratories of TROPOS.
For microscopic analysis, the protocol following Engel (2009) was applied. In short,
abundance, area, and size-frequency distribution of TEP were determined using a light
microscope (Zeiss Axio Scope A.1) connected to a camera (ColorView III). Filters were screened
at 200× magnification. About 10 pictures were taken randomly from each filter in two
perpendicular cross-sections (5 pictures each cross-section; dimension 2576 x 1932 pixel, 8-
bit color depth) and microscopic pictures of TEP in cloud water are shown in **Fig. 1**. Images
were then semi-automatically analyzed using ImageJ (Version 1.44). A minimum threshold
value of 16 μm² was set for particle size during particle analysis to remove the detection of
non-aggregate material by the program. This resulted in a minimum particle size of 4.5 μm
(assuming spherical particle).
***Insert Figure 1***
Blank filters were taken for aerosol sampling (inserting filters in the aerosol sampler
without probing them) and cloud water (filtering reagent water over a pre-cleaned filter),
stained and treated the same way as the microscopic analysis. Blank number concentrations
were on average 6% of the cloud water results and between 5% and 20% for aerosol results
and the blank values were subtracted from the samples.
The analysis of inorganic ions from $PM_{10}$ samples was performed with ion
chromatography and conductivity detection. Aqueous extracts of the aerosol samples were
made by ca. 25% of the $PM_{10}$ filter in 1.5 mL ultra-pure water (resistivity = 18.2 MΩ cm) for
one hour. After the filtration (0.45 μm syringe filter) of the extracts sodium ($Na^+$), calcium
($Ca^{2+}$), magnesium ($Mg^{2+}$), were analyzed by using ion chromatography (Dionex ICS-6000,
Thermo Scientific). The cations were separated in an isocratic mode (eluent: 36 mM
methanesulfonic acid) on a Dionex IonPac CS16-4μm column (2×250 mm) that was combined
with a Dionex IonPac CG16-4μm guard column (2×50 mm). The detection limits for the
determined ions were between 5 and 20 μg L$^{-1}$ (Zeppenfeld et al., 2021).
Non-sea-salt calcium was calculated from the ion ratio of $Ca^{2+}/Na^+$ in seawater of
0.038 (Turekian, 1968). Dust concentrations were estimated from the aerosol particle mass
concentrations as the residual mass after the subtraction of all analytical concentrations from
the $PM_{10}$ mass as described elsewhere (Fomba et al., 2014). Trace metal content was
determined using a Total Reflection X-Ray Fluorescence (TXRF) S2 PICOFOX (Bruker AXS,
Berlin, Germany) spectrometer equipped with a Molybdenum X-ray source (Fomba et al.,
2013). The cloud LWC was measured with a particle volume monitor (PVM-100, Gerber
Scientific, USA), which was mounted at the same height as the cloud water samplers.
INP number concentration ($N_{INP}$) were measured with two droplet freezing techniques
(LINA: Leipzig Ice Nucleation Array and INDA: Ice Nucleation Droplet Array) in different marine
compartments. The uncertainties of $N_{INP}$ are given by the 5% to 95% confidence interval and
the results are presented in (Gong et al., 2020a).
All the samples of this study are summarized in Table 1. In addition to samples from
the MarParCloud campaign, surface seawater samples obtained from the ETNA (Engel et al.
2020) were considered.
***Insert Table 1***
2.4 Enrichment factor





To determine enrichment or depletion of TEP in the atmosphere (i.e. on the aerosol
particles and in the cloud water) in relation to the TEP concentration in the ocean water, the
concept of the aerosol enrichment factor can be applied. To this end, the concentration of the
compound of interest in each compartment is related to the respective sodium mass
concentration, as sodium is regarded as a conservative sea salt tracer transferred to the
atmosphere in the process of bubble bursting (Sander et al., 2003). This concept is usually
applied for calculating the enrichment of a compound in the aerosol particles ($EF_{aer.}$) in relation
to seawater (Quinn et al., 2015), but was recently extended to calculate the enrichment of
organic compounds in cloud water ($EF_{cloud}$) in relation to seawater (Triesch et al., 2021a).
Therefore, in the following the enrichment factor is defined as $EF_{atm.}$ (atmosphere
enrichments factor) in equation 1.

$$EF_{atm.} = \frac{c\ (TEP)_{atm}/c\ (Na^+mass)_{atm}}{c\ (TEP)_{seawater}/c\ (Na^+\ mass)_{seawater}} \hspace{3cm} (1)$$

For equation (1), TEP number concentrations were converted to TEP volume
concentrations. To this end, for atmospheric and for oceanic samples, particle number
concentrations of TEP were extracted from the size distribution spectra and volume
concentrations were calculated (assuming spherical particles).  More detail on the conversion
can be found in the SI (Tab. S2-S5).

3 Results and Discussion
3.1 Concentration and size distribution of TEP
Within the three–weeks sampling period, TEP varied within one order of magnitude between
$7x10^2$ and $3x10^4$ #TEP m$^{-3}$ in the aerosol particles and between $4x10^6$ and $9x10^6$ #TEP L$^{-1}$ in the
cloud water (analysed diameter size range: ~ 4.5 to ~ 30 µm) as shown in **Fig. 2**. The cloud
water concentrations were converted to atmospheric concentrations using the measured LWC
of the cloud water (0.39 g m$^{-3}$) and resulted in concentrations of $2 – 4x10^3$ #TEP m$^{-3}$ (**Tab. S4**).
As a result, a striking similarity (agreement within one order of magnitude) for TEP
concentrations in the aerosol particles (average: $1x10^4$ #TEP m$^{-3}$, Tab. S2) and the cloud water
(average: $0.3x10^4$ #TEP m$^{-3}$, Tab. S4) was found, suggesting that the majority of the TEP
particles are activated to cloud droplets when a cloud forms.
***Insert Figure 2***

In addition, TEP were measured in four aerosol particle samples from the plunging
waterfall tank and the concentrations varied between $4x10^2$ and $3x10^3$ #TEP m$^{-3}$ (**Tab. S3**).
While the TEP concentrations in ambient aerosol particle and cloud water were not
significantly different (ANOVA, oneway, p = 0.054 at a 0.05 level), the tank-generated TEP





concentrations were significantly lower than the ambient aerosol TEP concentrations (ANOVA,
oneway, p = 0.004 at a 0.05 level). The TEP number concentrations measured in the different
atmospheric compartments, the ambient aerosol particles, the tank-generated aerosol
particles and the cloud water are summarized in **Fig. 3a** and the individual values are
presented in the **Tab. S2-S4**.
***Insert Figure 3***

Besides for the total number concentrations, TEP number size distribution were
derived from all ambient aerosol particle samples and are shown in **Fig. 4 (a-d)** in both, linear
and logarithmic form. In addition, the TEP number size distribution of one cloud water sample
is presented in **Fig. 4 (e, f).** All samples exhibited very similar trends in their size distribution,
with higher number concentrations for smaller sizes.
***Insert Figure 4***

From the observed size distributions, it can be assumed that the number
concentrations will continue to increase toward smaller sizes. A comparison of TEP number
concentrations in the ambient aerosol particles or cloud water to literature values is
challenging due to the availability of very few studies and different sample types and size
ranges regarded in different studies. However, the here observed trend in the TEP number
size distributions is consistent with studies from Kuznetsova et al. (2005) showing increased
TEP concentrations in simulated sea spray regarding particle sizes from 50 µm to 10 µm in
diameter. In addition, TEP mass concentrations showed a similar trend with higher
concentrations towards smaller particle sizes (size range 0.1-1 µm, Aller et al. (2017)), that
was, however not as pronounced as for TEP number concentrations observed here.
Regarding polymer gels in general, a strong increase with decreasing sizes was
observed for the polymer gels in cloud water in the high Arctic (north of 80°N) in late summer
using a very sensitive microscopic technique with epifluorescence (Orellana et al., 2011).
$2x10^9$ micrometer-sized polymer gels per $mL^{-1}$ and $2 – 6x10^{11}$ nanometer-sized polymer gels
per $mL^{-1}$ were observed and the majority of the particles were smaller than 100 nm (Orellana
et al., 2011). The measurements from Orellana et al. (2011) regarded a much smaller particle
diameter range (down to nm scale) compared to the present work and are therefore not
directly comparable. However, from the logarithmic TEP number concentration vs. diameter
relationship (**Fig.4**) we calculated TEP number concentrations for smaller particle ranges (sub-
micrometer size range). TEP number concentrations between $4.2x10^4$ #TEP $m^{-3}$ (low "TEP5"
case, equation from **Fig. 4b**) and $1.6x10^6$ #TEP $m^{-3}$ (high "TEP10" case, equation from **Fig. 4d**)
are calculated for $PM_1$ particles. The high but varying concentrations for the two cases
underlines the need for more measurements in the submicron range to derive robust
numbers. Similarly, a concentration of $3.0x10^8$ #TEP $L^{-1}$ for $PM_1$ particles in cloud water were



calculated and $2.1 \times 10^{10}$ #TEP $L^{-1}$ for $PM_{0.2}$ particles might exist in the submicron-size range
(following the equation from Fig. **4f**).

These calculations show that the number of gel-like particles in the high Arctic was still
several orders of magnitudes higher compared to TEP particles in the tropical Atlantic, e.g.
$10^{10}$ #TEP $L^{-1}$ (200 nm particles) in tropical cloud water observed here vs. $10^{11}$ #polymer gels
per $mL^{-1}$ ( $= 10^{14}$ #polymer gels per $L^{-1}$) from Orellana et al. (2011). If the TEP particles in the
tropical atmosphere comprise only a small subgroup of the total polymer gel number, or if the
total amount of gel-like particles is generally higher in Polar Regions remains to be
investigated.
3.2 Relating atmospheric TEP to the ocean
From a recent study of TEP number concentrations in different oceanic regions, TEP number
concentrations in surface waters (10 m depth) of the East Tropical North Atlantic (ETNA) were
obtained (Engel et al., 2020). ETNA is the region that geographically includes the Cape Verde
islands. The oceanic TEP number concentrations are shown in **Fig. 5** and are discussed in more
detail in Engel et al. (2020). The TEP in the ocean showed a similar size distribution compared
to the TEP in the atmosphere (i.e. aerosol particles and cloud water, **Fig. 4**) with increasing
TEP number concentrations toward smaller particle sizes (**Tab. S5** and more details in Engel et
al. (2020)).
***Insert Fig. 5***

A detailed comparison of #TEP in the ocean and in the atmosphere regarding the
identical size bins showed that the #TEP distribution among the different size bins were much
more balanced for seawater than for aerosol particles. In aerosol particles, on average 51% of
the #TEP were located in the smallest analysed size bin (4.5-7 μm) and show a sharp decrease
towards the second size bin (that contained 24% of the TEP) (Fig. 6). For the seawater TEP,
however, around 35% of the #TEP were found in the first size bin and the relative contribution
decreased uniformly towards the larger size bins (Fig. 6). This distribution is also visible in the
correlation curves of Fig 4 (b,d,f) and Figure 5b. The correlation curves for the aerosol particles
(and cloud water) have a steeper slope compared to the curve obtained for seawater TEP. This
could imply that i) the transfer of TEP from the ocean to the atmosphere is most efficient for
small size ranges, ii) larger TEP are converted to smaller TEP in the atmosphere (e.g. break
down), and /or iii) atmospheric in-situ formation mechanism of TEP preferably occur in smaller
particle size ranges. These considerations will be further evaluated in section 3.3.
***Insert Figure 6***

Ocean water, atmospheric particles, and cloud water are different marine
compartments and to compare seawater and atmospheric TEP concentrations in terms of





enrichment or depletion, the atmospheric enrichment factor $EF_{atm.}$ (Equation 1) was
calculated. In order to compare the same TEP diameters in all compartments, the size range
between 5 µm (lower limit for atmospheric measurements) and 10 µm (typical upper limit for
ambient aerosol particles) was regarded and converted from number to volume concentration
(more details in Table S2-S4 and Fig. S1). For ocean water, TEP number concentrations of
$3.5x10^3$ #TEP mL$^{-1}$ (= $3.5x10^6$ #TEP L$^{-1}$) and a TEP volume concentration of $4.6x10^5$ µm$^3$ TEP mL$^{-1}$
$^1$ (= $3.5x10^8$ µm$^3$ TEP L$^{-1}$) were obtained. The respective values for the TEP volume
concentration of ambient and tank-generated aerosol particles, as well as for the cloud water
are listed in Tables S2-S4 and illustrated in **Fig 3b**. The factors given here are subject to some
uncertainties and represent lower limits. An error discussion is introduced in the Supporting
Information as an appendix to Table S2. It is clearly visible that the $EF_{aer. ambient}$ are significantly
higher than the $EF_{aer. tank}$ (ANOVA, oneway, p = 0.0017 at a 0.05 level) with average values of
$9x10^3$ and 50, respectively. The average $EF_{cloud}$ was $1x10^3$. This means that the enrichment of
TEP derived from the plunging waterfall tank, representing the bubble-bursting transfer, is
about two orders of magnitude lower compared to the enrichment of TEP in the ambient
aerosol particles. In the following, this finding will be discussed in more detail considering
studies available from literature.

Atmospheric enrichment of ocean-derived OM, have often been reported (e.g.
Facchini et al., 2008;Keene et al., 2007;O'Dowd et al., 2004;Schmitt-Kopplin et al.,
2012;Triesch et al., 2021a;Triesch et al., 2021b;van Pinxteren et al., 2017). Submicron particles
are usually strongly enriched with organic matter with aerosol enrichment factors $EF_{aer.}$ of $10^3$
up to $10^5$ (Quinn et al., 2015 and references therein). The enrichment in supermicron aerosol
particles is, however, significantly lower. Laboratory studies showed enrichment of OM in the
order of $10^2$ (Hoffman and Duce, 1976;Keene et al., 2007;Quinn et al., 2015). From the
MarParCloud campaign, enrichment factors of free amino acids were between 10 and 30 in
ambient supermicron particles (Triesch et al. 2021a). Kuznetsova et al. (2005) reported TEP
enrichments in freshly produced sea spray with $EF_{aer.}$ = 44 ± 22 based on TEP number
concentration. Consequently the here reported $EF_{aer. tank}$ (50 ± 35) are well in-line with
published enrichment factors for OM in general and TEP specifically. However, the $EF_{aer. ambient}$
($9x10^3$) were orders of magnitude higher than reported enrichment factors for supermicron
aerosol particles. Enrichment factors of OM in cloud water are hardly available; we recently
reported an enrichment of $10^3 – 10^4$ of free amino acids in cloud water from the MarParCloud
campaign (Triesch et al., 2021a) that were higher than the here observed $EF_{cloud}$.

The concept of the aerosol enrichment factor originally originates from controlled tank
experiments where a direct transfer of compounds from the ocean via sea-spray aerosol
formation occurs. Obviously, this does not automatically correspond to the ambient
environment as mixing processes, aging, and further transformation reactions are not
accounted for. However, the $EF_{aer. ambient}$ which is much bigger than $EF_{aer}$ , $EF_{aer. tank}$  and the
comparison of $EF_{cloud}$ towards former literature data clearly show the presence of significantly





more TEP in ambient aerosol and cloud water compared to oceanic seawater which will be
discussed in detail  in the following section.

3.3 Possible sources and atmospheric formation pathways of TEP

3.3.1 Primary TEP sources

The high abundance of TEP in the aerosol particles and cloud water might correspond
to an oceanic transfer within the process of bubble bursting. To investigate a linkage to the
bubble bursting transfer, TEP concentrations were correlated to the sea-spay tracers sodium
and magnesium. To account for biases due to a number-based (TEP) and mass-based (sodium,
magnesium) comparison, the particle volume of TEP was calculated from the particle number
concentrations (regarding the size range: 5-10 µm). To this end, from each particle diameter
within a size range of 5-10 µm, the respective volume was determined, assuming spherical
particles, and summed up (data in **Tab. S2**). This transformation accounts for the fact that big
TEP particles likely possess a large mass but a low number concentration and vice versa.
Reasonably good correlations of TEP to sodium and magnesium, ($R^2$ = 0.5, **Fig. 7a,b**)
suggested some connection to a bubble bursting transfer. This was further supported by a
moderate correlation of $R^2$ = 0.5 of TEP to sea-salt calcium (Ca$_{ss,}$ **Fig. 7c**), which was absent for
non-sea-salt calcium and total calcium (**Fig. 7d**).

***Insert Figure 7***


Despite this correlation of TEP to sea spray tracers, the high abundance and
enrichment of #TEP in the ambient aerosol particles compared to literature data (Kuznetsova
et al., 2005) and compared to the concentration and enrichment of the #TEP from the plunging
waterfall tank performed here, suggests that additional TEP sources in the ambient
atmosphere exist from which TEPs are added to their primary transfer by bubble bursting from
the oceans. At the Cape Verde islands, besides the ocean, mineral dust is an important aerosol
particle source (Fomba et al., 2014). TEP are generally attributed to be ocean-derived
compounds however, dust has often been reported to transport attached biological particles
(Maki et al., 2019;Marone et al., 2020). During the MarParCloud campaign, dust influences
were low to moderate and the aerosol particle mass was found to be predominantly of marine
origin (Fomba et al., 2014;van Pinxteren et al., 2020). Some dust influences were visible
though, e.g. variations in the particle number concentrations, with elevated concentrations
on (even low) dust influenced air masses (Gong et al., 2020b). TEP number concentrations
showed no clear connection to the ambient dust concentrations (**Fig. 2**). Within periods of
moderate dust, TEP were partly below the detection limits (on 26.09.2017) and partly
exhibited high concentrations (e.g. on 28. and 29.09.2017). A correlation between TEP and
dust was not found ($R^2$ = 0.05, **Fig. 7e**) therefore, we do not consider dust to be a transport



medium for TEP to the particles or cloud water. However, dust might play a role in abiotic TEP
formation, as discussed in chapter 3.3.2.1.
3.3.2. In-situ formation
3.3.2.1 Abiotic formation
In aquatic environments, abiotic TEP formation has been reported to happen via
several pathways, including spontaneous assembly from TEP precursors (Passow, 2002b). The
aerosol particle and cloud water samples from the MarParCloud campaign investigated here
showed high mass concentrations of amino acids (up to 6.3 ng m$^{-3}$ in the submicron aerosol
particles and up to 490 ng m$^{-3}$ in the cloud water, published in Triesch et al.  (2021a)) as well
as dissolved polysaccharides (up to 2 ng m$^{-3}$ in the submicron aerosol particles and up to 2400
ng m$^{-3}$ in the cloud water, results in preparation for publication). In the ocean, the dissolved
polysaccharides are known TEP precursors (Passow, 2002b) and Wurl et al.  (2011) determined
abiotic TEP formation rates from dissolved polysaccharide concentration in various oceans.
Therefore, spontaneous TEP formation from the (high) abundant dissolved polysaccharides
likely contributed to the high TEP concentrations observed in the ambient atmosphere in the
present study.
Another important parameter likely impacting TEP formation is the presence of
mineral dust. As already discussed above, dust mass concentrations were low to moderate,
however not negligible, during the MarParCloud campaign. In laboratory minicosm studies,
the addition of dust to oceanic water resulted in an acceleration of the kinetics of TEP
formation leading to the formation of fast sinking particles (Louis et al., 2017). This process
likely happens due to particle aggregation, meaning that dissolved OM and dust aggregate to
form TEP (Louis et al., 2017). In addition, dust particles in cloud water might promote
turbulence, which, in aquatic media, has been suggested to enhance abiotic TEP formation
(Passow, 2002b). The dust deposition at the Cape Verdes has been recognized as a potentially
large contributing factor to the TEP enrichment in the SML at the Cape Verdes (Robinson et
al., 2019a). Here, we speculate that even low concentrations of mineral dust can influence the
TEP formation on the aerosol particles and in the cloud water. This is further supported by the
microscopic detection of dust in the cloud water (**Fig. 1**), that likely enhance the possibility
that particles in the cloud water collide and stick. Consequently, while dust did not seem to
serve as a transport medium for TEP (see sec. 3.3.1), dust may contribute to in-situ TEP
formation in cloud water due to abiotic particle aggregation.
From atmospheric studies, marine gel particles have been reported to undergo a
volume phase transition in response to environmental stimuli, such as pH and temperature as
well as cleavage of their polymers due to UV radiation (Orellana et al., 2011). UV radiation can
break down microgels in the ocean to a high number of smaller (nano-sized) particles (Orellana
and Verdugo, 2003) – a mechanism that is expected highly relevant in the atmosphere where



UV radiation is higher than in seawater. Furthermore, it has been shown that a lowering of
the pH from neutral conditions (7 or 8) to 4.5 causes a sudden transition of gel particles in
which the polymer network collapsed to a dense, non-porous array (Chin et al., 1998). The pH
in the cloud water analysed here was between 6.3 and 6.6. As TEP are reported to exhibit a
gel-like character (Passow, 2002b), volume and number concentrations might be affected by
the different pH, ion density, temperature and pressure in the atmosphere. At cloud water
pH-values such as measured here, marine gels have been found to split into smaller units (Chin
et al., 1998), that are below the minimum detectable particle size of 4.5 μm. This could explain
the lower concentrations in cloud water ($2 - 4x10^3$ #TEP m$^{-3}$) compared to ambient aerosol
particles ($7x10^2 - 3x10^4$ #TEP m$^{-3}$). Hence, the different environmental stimuli likely impact
atmospheric TEP formation and might lead to the formation of smaller particles. However,
from our data we cannot report on the quantity of these effects and such investigations
warrant further studies.
3.3.2.2 Biotic formation
Besides abiotic pathways, in aqueous media, TEP can be directly released as
particulates from aquatic organisms involving phytoplankton and bacteria (Passow, 2002a)
Biotic TEP formation has by now been studied for seawater and lakes (Passow, 2002a)
however, bacteria are also present in the atmosphere and likely transferred from the ocean
via sea spray (Rastelli et al., 2017) and can survive in cloud droplets (Deguillaume and al.,
2020). The bacterial abundance in cloud water samples taken at Mt. Verde during the
MarParCloud campaign ranged between 0.4 and $1.5x10^5$ cells mL$^{-1}$ (van Pinxteren et al., 2020).
This concentration is one to two orders of magnitude higher than the TEP concentrations. The
bacterial tracer muramic acid (Mimura and Romano, 1985) was detected in the aerosol
particles and cloud water sampled here in considerable concentrations (~ 25 nM, data not
shown), strongly suggesting bacterial activity in cloud water. We cannot derive conclusions on
the origin of the bacteria measured in cloud water reported here, however the transfer of
bacteria from the ocean to the atmosphere has been shown before (Rastelli et al.,
2017;Uetake et al., 2020). TEP are known to be closely connected to bacteria in different ways
(Passow, 2002b;Passow, 2002a), therefore, the presence of bacteria in the atmosphere
exhibits a potential source of cloud water TEP observed here. Furthermore, TEP are strongly
colonialized by bacteria (Busch et al., 2017;Zäncker et al., 2019). Hence, TEP can be a transfer
vector for bacteria from the ocean to the atmosphere and/or act as a medium for bacterial
colonisation in marine clouds.
The presence of active enzymes on ambient aerosol particles (enriched compared to
seawater) and therefore biogenic in-situ cycling of OM through enzymatic reactions in
atmospheric particles was recently suggested (Malfatti et al., 2019). This is well in-line with
the findings that the aerosol particles and cloud water from the MarParCloud campaign
contained high concentrations of OM (amino acids, lipids), assumingly connected to the
biogenic formation (Triesch et al., 2021a;Triesch et al., 2021b). A combined approach of



laboratory experiments and modelling recently underlined the importance of biotic (and
abiotic) formation processes of OM in clouds (Jaber et al., 2021).
Considering recent literature and the data reported here, we suggest that in-situ TEP
formation related to biogenic processes and likely connected to bacteria, as reported for
seawater, are important in the marine atmosphere as well.
3.4 Connecting TEP and Ice nucleating particles (INP)
Different kinds of ice-nucleating macromolecules have been found in a certain range
of biological species and consist of a variety of chemical structures including proteins,
polysaccharides (Pummer et al., 2015) and lipids (DeMott et al., 2018). TEP, consisting of
polysaccharidic chains, bridged with divalent cations, may therefore possess good properties
to act as INP, however, such a link has not yet been shown in field experiments.
During the MarParCloud campaign INP number concentration ($N_{INP}$) was measured in
different marine compartments and the results are presented in Gong et al. (2020a). By
combining INP concentration in the seawater, aerosol particles and cloud water, it was found
that $N_{INP}$ in the atmosphere were at least four orders of magnitude higher than what would
be expected if all airborne INP would originate from sea spray. The measurements indicated
that other sources besides the ocean, such as mineral dust or other long-ranged transported
particles, contributed to the local INP concentration. However, some indications for
contributions of biological particles to the INP population were obtained (details in Gong et
al., 2020a). Nevertheless, the sources of INP could not be revealed in detail.
In the present study, quantitative INP data (presented in Gong et al. 2020a) and TEP
data measured from the same campaign were compared. To this end, INP concentrations
achieved from $PM_{10}$ quartz-fiber filters taken at the CVAO during the same period as the TSP
filters were compared with the TEP measurements. In addition, cloud water INP and TEP data
obtained from the same samples were combined.
TEP number concentrations were on average between $10^3 – 10^4$ m$^{-3}$ in the ambient
aerosol particles, whereas INP number concentrations at -15 °C were between $10 – 10^2$ m$^{-3}$
(Gong et al., 2020a). It is interesting to note that the TEP concentrations in the ambient aerosol
particles were about two orders of magnitude higher compared to INP concentrations. Similar
findings were obtained for the cloud water comparisons; TEP concentrations (~ $10^6$ L$^{-1}$) were
on average two orders of magnitude higher than INP number concentrations at -15 °C in cloud
water (~ $10^4$ L$^{-1}$) (Gong et al., 2020a).
The correlation between INP (active at -15°C) and TEP concentrations was weak with
$R^2$ = 0.3 (**Fig. 5f**), showing that a direct link between INP and the entire TEP number
concentrations was not very pronounced. It needs to be underlined that TEP concentrations
below a particle size of 4.5 µm are not included here and according to the size distribution,
the TEP concentrations are increasing towards smaller sizes. Most of the here reported TEP
particles were in the supermicron sizes (~ 4.5 – 14 µm, **Fig. 4**). However, the biologically active
$N_{INP}$ at the Cape Verdes were mainly present in the supermicron mode (> 1 µm) (Gong et al.,





615 2020a), hence a comparison with the TEP particle concentrations above 5 µm seems justified.
616 Nevertheless, future studies should concentrate on the exact same size ranges for TEP and
617 INP.

618  The INP functionalities of biomolecules are not straightforward and whether a
619 macromolecule acts as INP is depending on many factors, as its size, proper position of
620 functional groups, and their allocation (Pummer et al., 2015). Typically, not the entire surface
621 of an INP but rather specific areas (active sites) participates in ice nucleation. This means that
622 despite TEP likely providing INP properties, only a fraction of TEP, if any, might be able to act
623 as INP. This hypothesis is supported by the findings that marine gels exhibit hydrophobic and
624 hydrophilic surface-active segments, strongly suggesting a dichotomous, non-uniform
625 behaviour of polymer gels (Leck et al., 2013;Orellana et al., 2011;Ovadnevaite et al., 2011). As
626 mentioned in 3.3.2.1 and 3.3.2.2, TEP are often attached to, or colonized with bacteria.
627 Bacteria itself, have been shown to provide excellent INP functionalities (Pandey et al., 2016)
628 and TEP might act as a carrying medium for INP, such as bacteria. Bacteria concentrations
629 were higher than TEP concentrations and also higher than INP concentrations. However, only
630 a fraction of all bacteria (0.5 – 25%) is associated with TEP and, vice versa, not all TEP are
631 colonized by bacteria (Passow, 2002b). There is an indication that especially in oligotrophic
632 waters, as are the Cape Verde islands, the fraction of bacteria attached to TEP is comparably
633 low (Schuster and Herndl, 1995). Hence, the concentration range of bacteria-colonized TEP in
634 relation to INP is worth further consideration. This might help to unravel if a functional
635 relationship between bacteria-colonized TEP and INP exists and if a certain part of TEP contain
636 fragments in the biological INP population that, beyond dust, play a role in the Cape Verde
637 atmosphere.

639 4 Conclusion

641 This study presented TEP number concentrations > 4.5 µm in ambient atmospheric samples
642 from the tropical Atlantic Ocean during the MarParCloud campaign as well as in generated
643 atmospheric particles using a plunging waterfall tank. The atmospheric TEP showed a similar
644 size distribution compared to the TEP in the ocean with increasing TEP number concentrations
645 toward smaller particle sizes, however the #TEP distribution among the different size bins
646 were much more balanced for seawater than for aerosol particles where half of the #TEP were
647 located in the smallest analysed size bin (4.5-7 µm). Based on $Na^+$ concentrations in sea water
648 and the atmosphere, the enrichment of TEP in the tank generated aerosol particles was well
649 in-line with another study. The TEP enrichments in the ambient atmosphere were, however,
650 up to two orders of magnitude higher compared to the tank study and such high values are
651 thus far not reported for supermicron aerosol particles. We speculate that the high
652 enrichment of TEP in supermicron particles and in cloud water result from a combination of
653 enrichment during bubble-bursting transfer from the ocean and in-situ atmospheric
654 formation. We propose that similar (biotic and abiotic) formation mechanism reported for TEP



formation in the (sea)water might take place in the atmosphere as well, as the required
conditions (e.g. high concentrations of dissolved TEP precursors such as polysaccharides,
presence of bacteria in the cloud water) were given. An assessment of the importance of the
biotic versus the abiotic TEP formation pathways in the atmosphere, however, needs further
investigations. TEP concentrations in the atmosphere were two orders of magnitude higher
than INP concentrations in the aerosol particles and cloud water, respectively. However, only
a part of the TEP population, assumingly the one colonized by bacteria, might contribute to
INP population, and are worth further studies. Finally, while dust might be a dominant INP
source in the here investigated tropical Atlantic region close to the Saharan desert, in other
remote oceanic locations, marine gel particles, their in-cloud formation and connection to
bacteria in the atmosphere could be highly relevant for a better understanding of marine
cloud properties.

Data availability
The data are currently made available through the World Data Centre PANGAEA and the link
will be included in the next version of the manuscript.  INP concentrations are accessible under
the following link: https://doi.pangaea.de/10.1594/PANGAEA.906946.
Special issue statement
Acknowledgement
We acknowledge the funding by the Leibniz Association SAW in the project "Marine biological
production, organic aerosol particles and marine clouds: a Process Chain (MarParCloud)"
(SAW-2016-TROPOS-2), the Research and Innovation Staff Exchange EU project MARSU
(69089) and the Deutsche Forschungsgemeinschaft (DFG, German Research Foundation) –
Projektnummer 268020496 – TRR 172, within the Transregional Collaborative Research
Center "ArctiC Amplification: Climate Relevant Atmospheric and SurfaCe Processes, and
Feedback Mechanisms (AC)[3]" in sub-projects B04. We thank the CVAO site manager Luis Neves
as well as René Rabe and Susanne Fuchs for technical and laboratory assistance. We further
acknowledge the professional support provided by the Ocean Science Centre Mindelo (OSCM)
and the Instituto do Mar (IMar).

Author contributions
MvP led the MarParCloud campaign and, together with the campaign participants KWF, XG,
EB, NT, BR, FS and HW performed the aerosol particle and could water sampling at the Cape
Verde island. EB designed and operated the plunging waterfall tank. BR performed the
microscopic TEP measurements and XG made the INP analysis. AE contributed the seawater
TEP data. MvP performed the data interpretation with help from SZ and BR. MvP wrote the
manuscript with contributions from all authors.



Competing interest
The authors declare that they have no conflict of interest.

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





**Caption of Figures:**

**Figure 1**: Microscopic analysis of TEP from the cloud water sample "WW5" (sampling interval: 28.09. 19:30 – 29.09. 7:30 local time). Blue particles are TEP, stained with Alcian Blue solution; brownish particles in the right picture are assumingly dust particles. The scale refers to 50 µm.

**Figure 2:** TEP number concentrations in the aerosol particles (red bars) and in the three cloud water samples (blue-red squares). TEP concentrations were below the limit of detection (LOD) on 26th and 27th of September 2017. The backgrounds represent the dust classification according to the ambient dust concentrations (blue: dust < 5 µg m$^{-3}$ marine conditions; yellow: dust < 20 µg m$^{-3}$ (low dust); brown: dust < 60 µg m$^{-3}$ (moderate dust). From underlined dates (22.09 -> "TEP5" and 28.09.2017 -> "TEP10") TEP number size distributions were measured.

**Figure 3**: Box and whisker plot of the TEP number concentrations (a) and the enrichment factors (b) in the ambient (n=18) and tank-generated (n=4) aerosol particles and in the cloud water samples (n=3), Each box encloses 50% of the data with the mean value represented as an open square and the median value represented as a line. The bottom of the box marks the 25% limit of the data, while the top marks the 75% limit. The lines extending from the top and bottom of each box are the 5% and 95% percentiles within the data set, while the asterisks indicate the data points lying outside of this range ("outliers").

**Figure 4**: TEP number size distribution in the aerosol particles and cloud water in linear and logarithmic form; panels (a) and (b) show aerosol particle sample "TEP 5" (sampling start: 22.09.2017), panels (c) and (d) show aerosol sample "TEP 10" (sampling start: 28.09.2017), panels (e) and (f) show cloud water sample "WW5" (sampling interval: 28.09. 19:30 – 29.09. 7:30 local time. The lower limit of the resolution of the microscope was 16 µm$^2$ resulting in a particle diameter of 4.5 µm (assuming spherical particle). Each bar in a), c), and e) represents the summed up particle number concentrations (within 1.5 µm), e.g. the first column represents the summed up concentrations between 4.5 and 6 µm.

**Figure 5:** TEP number size distributions in the ocean surface water (sampling depth: 10 m) from the East Tropical North Atlantic (ETNA), averaged over three stations from Engel et al (2020). The data in this Figure show the size distribution between ~ 5 and ~ 30 µm, matching the investigated aerosol size range (**Fig. 4**). The whole size spectrum is shown in **Tab. S5.**

**Figure 6:** Relative contribution of the TEP number concentrations in the aerosol particles (left) and in the ocean surface water (right) regarding the identical size bins.



**Figure 7:** Correlations of TEP volume concentrations (size range: 5-10 µm) to chemical
parameters (inorganic constituents PM10) and dust (PM10), as well as correlation of TEP
number concentration and INP number concentrations. Inorganic constituents were
measured with ion chromatography and dust concentrations were derived from PM10
concentrations as reported elsewhere (Fomba et al., 2013;van Pinxteren et al., 2020).
Measurements of INP number concentrations and error bars are explained in (Gong et al.,
2020a)





Table 1. Overview of sampling locations, types and measurements

| Sampling site | Campaign | Sample type | Coordinates | No. of samples | Measurements (Particle sizes) |
|---|---|---|---|---|---|
| CVAO | MarParCloud 2017 | Ambient aerosol particles Inlet hight: 42 m a.s.l | 16° 51.49´ N, 24° 52.02´ W | 20 20 | #TEP (TSP) Inorganic ions (PM$_{10}$) |
| Mt- Verde | MarParCloud 2017 | Ambient cloud water Inlet hight: 746 m a.s.l | 16°52.11ˈN, 24°56.02ˈW | 3 | #TEP Inorganic ions |
| Plunging waterfall tank (operated at CVAO) | MarParCloud 2017 | Tank-generated aerosol particles | 16° 51.49´ N, 24° 52.02´ W | 4 | #TEP (TSP) Inorganic ions (TSP) |
| ETNA (Mauretanian upwelling) | M107 RV Meteor 2012 | Ocean surface water | 18.00/18.19´N -16.50/72.02´E | 6 | #TEP |










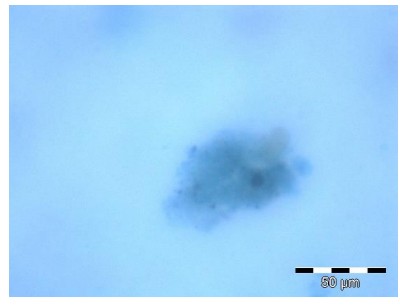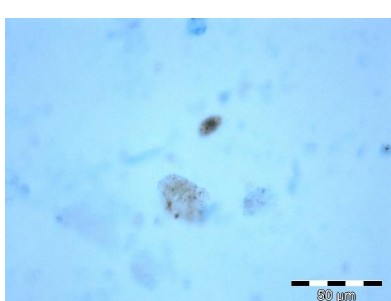




1061                                                                              Figure 1










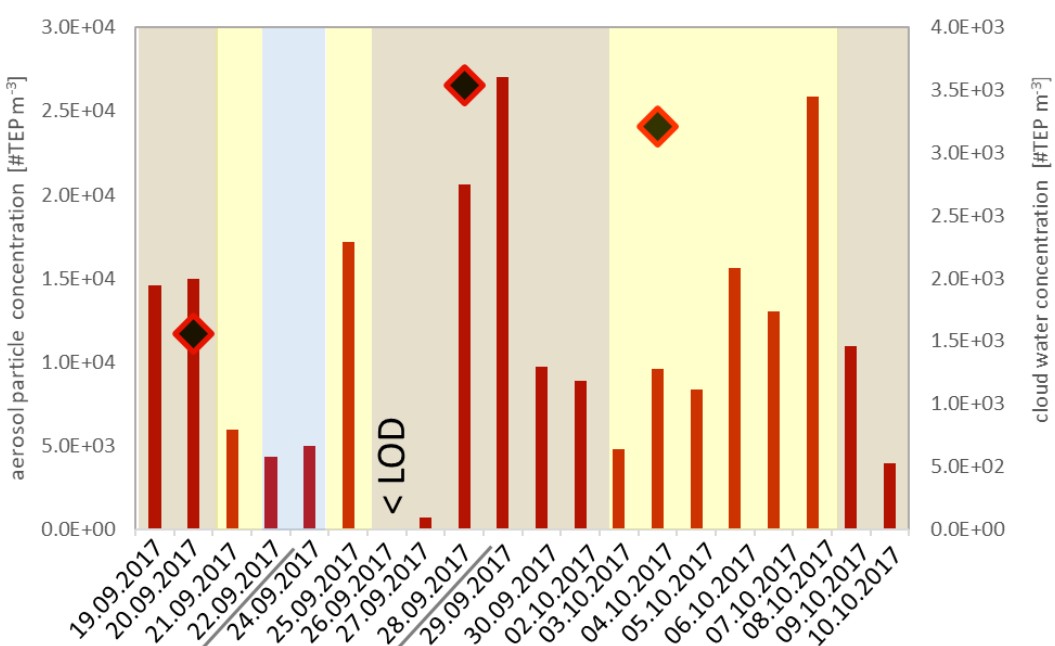

1068                                                                                                  Figure 2













(a)


(b)


Figure 3




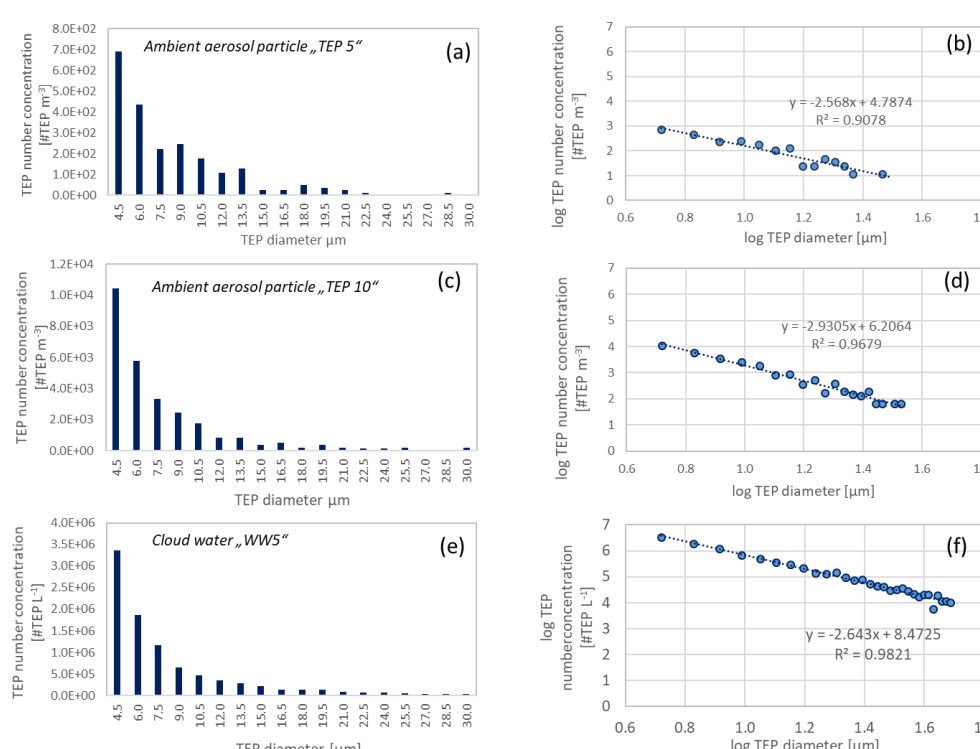


1091                                                                                        Figure 4










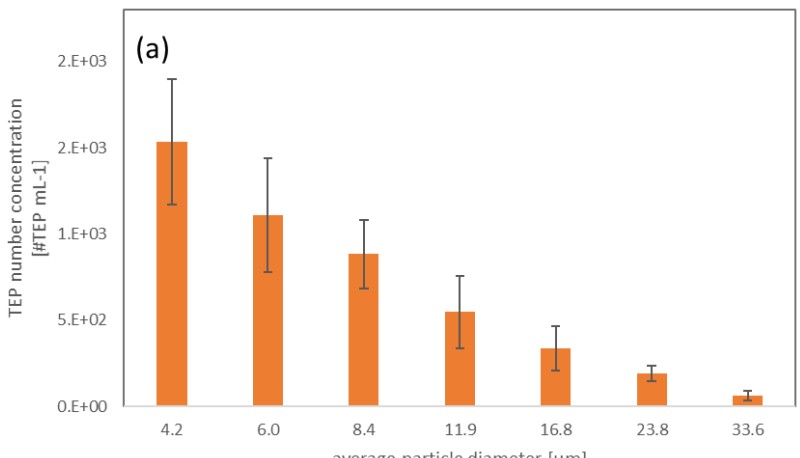











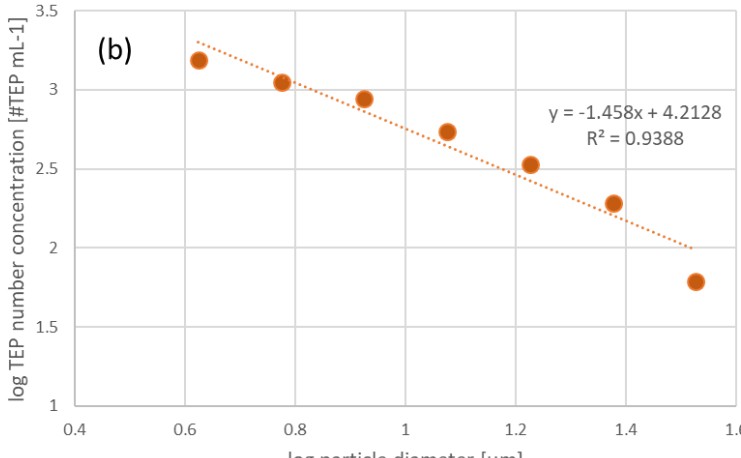


Figure 5











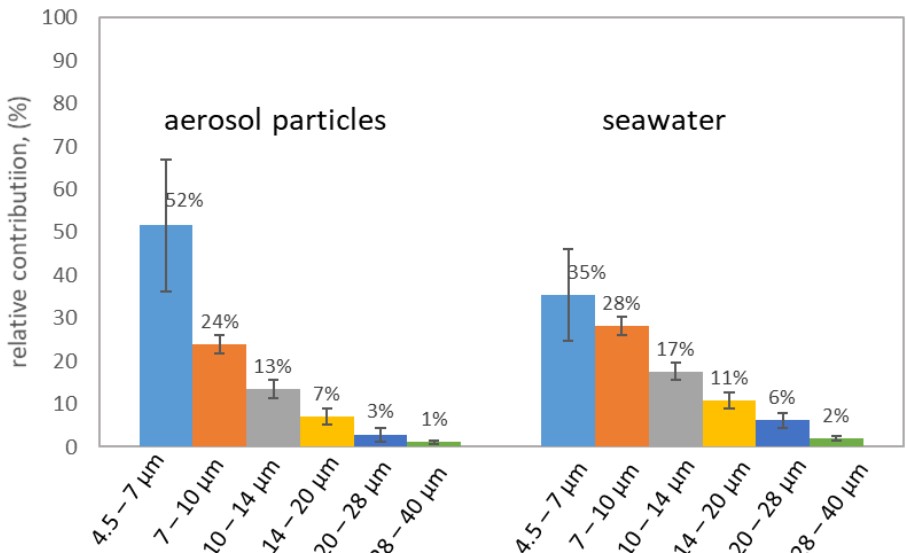






1121                                                                                      Figure 6




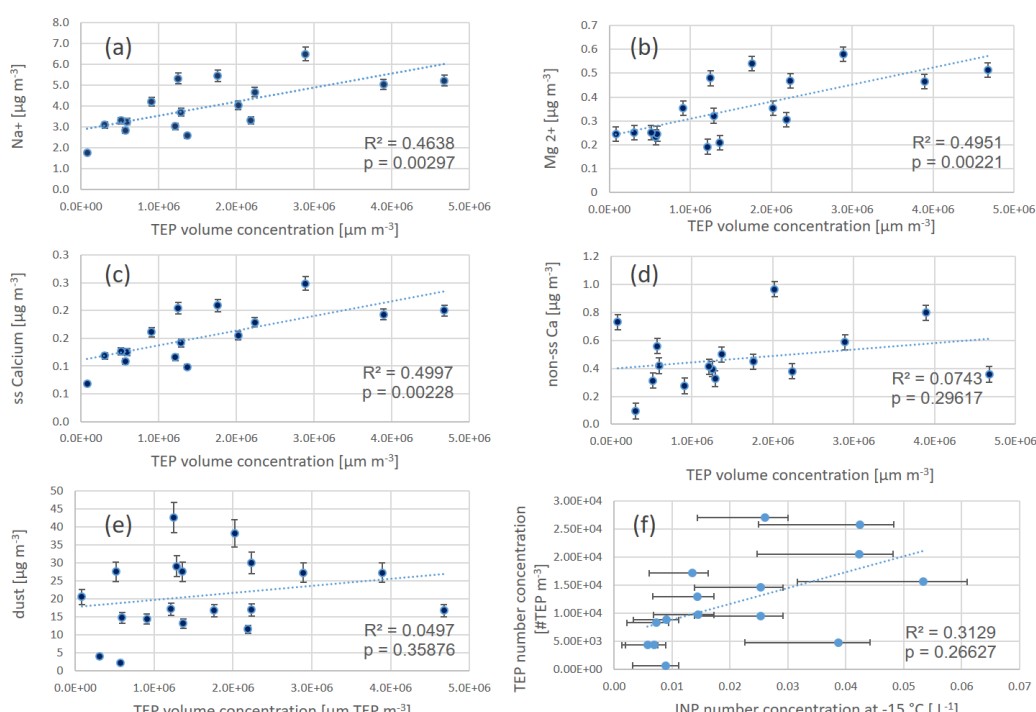


1123                                                              Figure 7

