# Peer review of "High number concentrations of transparent exopolymer particles (TEP) in ambient aerosol particles and cloud water - A case study at the tropical Atlantic Ocean Manuela van Pinxteren1, Tiera-Brandy Robinson2, Sebastian Zeppenfeld1, Xianda Gong3+, Enno B"

_Atmospheric Chemistry and Physics, 2021_

## Author Comment (AC1)

Referee 1:

Transparent exopolymer particles (TEP) have been shown as highly prevalent in the sea surface microlayer (SML) with a potentially significant effect on air-sea release of marine aerosols. They are also highly important in sedimentation processes and carbon cycling in the sea. This study presents TEP number concentrations > 4.5 μm in ambient atmospheric samples from the tropical Atlantic Ocean during the MarParCloud campaign as well as in generated atmospheric particles using a plunging waterfall tank. The publication present a robust data set on atmospheric TEP measurements that are rare to date and concluded interesting new findings. Authors speculate that the high enrichment of TEP in supermicron particles and in cloud water result from a combination of enrichment during bubble-bursting transfer from the ocean and in-situ atmospheric formation. They also propose that similar (biotic and abiotic) formation mechanism reported for TEP formation in the (sea)water might take place in the atmosphere as well, as the required conditions (e.g. high concentrations of dissolved TEP precursors such as polysaccharides, presence of bacteria in the cloud water) were given. I believe that this manuscript is suitable for ACP (and notably this Special Issue). The publication sets new research trends and points to the necessity of further investigations on the assessment of the importance of the biotic versus the abiotic TEP formation pathways in the atmosphere. It seems to be important to continue such a measurements in other remote oceanic locations, since marine gel particles, their in-cloud formation and connection to bacteria in the atmosphere could be highly relevant for a better understanding of marine cloud properties. Especially important, in my opinion, is the determination of TEP concentrations in waters with high productivity, i.e. the Baltic Sea, especially since high TEP concentrations are usually associated with phytoplankton blooms, with the special importance of diatoms.

We thank the Reviewer for the evaluation and the constructive comments. Replies to the specific Referee's comments are provided below in red and new parts included in the manuscript are marked in **green**. Line numbers refer to the revised (clean) version.

1. **The introduction** is written very clearly and allows to fully understand the mechanisms of TEP formation, their properties, distribution to the atmosphere and the role they play in the environment. This chapter describes the current state of knowledge on TEP subject.
   **Please indicate** some examples of particles or highly dense matter, that support downward carbon fluxes and those, that will result lead to rise of TEP to the surface and to form or stabilize the SML (Lines 104-108).

As the reviewer suggested, we included some more references that show downward carbon fluxes and the rise of TEP to the surface forming the SML. We included the study of (Logan et al., 1995) who studied the "rapid formation and sedimentation of large aggregates is predictable from coagulation rates (half-lives) of transparent exopolymer particles (TEP)" as well as the review paper of (Mari et al., 2017)" Transparent exopolymer particles: Effects on carbon cycling in the ocean" which covers all aspects of

TEP rising and sinking. In addition, we included two papers studying TEP at the SML (Wurl and Holmes, 2008;Wurl et al., 2011).

2. Line 101- there is no indentation in the paragraph

This was corrected.

**Material and methods**

3. Lines 2015-216; 223 and 235- what acid was used? Could the use of the acid influence further analysis? How were filters/bottles etc., treated after using acid? - there is no precise description or reference to the literature in which it was previously described. The remaining methodological description does not raise my reservations.

The revised version described the cleaning procedure in more detail. It is a standard procedure for DOC and TEP analysis and after cleaning with HCl, the equipment is rinsed with ultrapure water as recommended in Engel et al. (2009). This cleaning procedure removes contaminations very efficiently and the usage of acid does not influence further analysis. We added a reference and it reads now:

Line 227-230: "All equipment that was in contact with the cloud water samples (Teflon®strands, sampling bottles, filters) had been cleaned with 10% HCl and rinsed with ultrapure water (resistivity=18.2MΩ cm) before each application as recommended in (Engel, 2009)".

And:

Line 201-203: "The PC filters had been cleaned with 10% HCl and rinsed ultrapure water (resistivity = 18.2MΩ cm) water before application."

**Results and Discussion**

The discussion is logical and brings a lot of interesting information. Statistical analysis of the results also does not raise any reservations. Below are some comments, questions and suggestions.

4. Lines 330-333 - It seems to me that it is exaggerated to say that the majority of the TEP particles are activated to cloud droplets when a cloud forms, only on the basis that striking similarity for TEP concentrations in the aerosol particles and the cloud water was found Especially since the samples from the clouds were collected only in the amount of 3 ... - Please explain where this statement came from.

We thank the reviewer for this comment. This was clearly a mistake and the term "the majority of the TEP are activated to cloud droplets" was not correct. Unfortunately, the cloud water sampling cases for TEP measurements were limited. For a more correct assignment, we compared the #TEP concentrations in cloud water to the #TEP concentrations in the ambient aerosol particles when sampling times coincided (20.09., 28.09., and 4.10. 2017). For these dates, the cloud water #TEP concentrations made up between 10 and 34% (average: 20%) of the #TEP ambient aerosol concentrations. In addition, we related the average cloud water #TEP concentrations to the average #TEP concentrations in the ambient aerosol, which showed that the cloud water #TEP concentrations made up 24% of the #TEP ambient aerosol concentrations. Regarding these numbers, we conclude that the #TEP concentrations in cloud water were about 20% of the #TEP concentrations in the ambient aerosol (with a good agreement regarding the matching dates and the average values) and added this information in the manuscript as follows:

Line 336-338: "Comparing the #TEP concentrations in cloud water to the ones in the ambient aerosol particles suggested that about 20% of the ambient TEP particles are activated to cloud droplets when a cloud forms."

5. Line 343- I propose to divide Fig. 3 and here leave only part "a", because in the text there is description only of that figure. Part b I suggests inserting after line 433- when the authors describe the EF coefficients.

We understand the point the reviewer raised and it makes sense from a logical point of view. However, as the Figure 3b is in a very similar format as Figure 3a they somehow belong together (although being discussed in different chapters). For the sake of clarity as well as for avoiding including too many single Figures, the authors would prefer to leave it as it is.

6. Line 365- remove "polymer gels"

Done

7. Lines 391-392- Remove "and are discussed in more detail in Engel et al. (2020)". There is a reference to this literature at the end of the paragraph, which is enough.

Done

8. Lines 415-418- Remove "Ocean water, atmospheric particles, and cloud water are different marine compartments". It doesn't sound logical. I propose to start the sentence with: "To compare seawater and atmospheric TEP concentrations in terms of...".

We agree and adjusted the text as suggested.

9. Lines 479-484- I. I think that an important aspect of the influence of wind speed on the generation of marine aerosols has been overlooked in this section and should be noted. However, there is a mention of this on lines 99-101 (Introduction). Wind speed has a direct impact on the concentration of sea salt (both sodium and chloride) in the atmosphere above the sea/ocean. The effectiveness of marine drops generating and dispersing of large sea salt nuclei from the surface of breaking waves increases with square of the wind speed and, in the case of whitecaps occurrence (wind speed above 10 m s$^{-1}$), changes with wind speed cubed (Nair et al, 2005; O'Dowd and Hoffmann, 2005).

We thank the reviewer for this comment. The wind speed data are listed in the SI (Tab. S1) and present average data of a 24 h sampling interval. As shown in the manuscript, we observed a good correlation of sodium, magnesium and sea-salt calcium to the TEP concentrations that indicates some connection to bubble bursting transfer. However, there is no correlation between #TEP and wind speed. In addition, the correlation between sodium and wind speed is surprisingly weak (R$^2$ = 0.2). It may be that since wind speed data represented an average value of 24 hours, short but pronounced changes in the wind speed were not visible in the average wind speed value. We think this is a separate topic to investigate.

We added to the manuscript: Line 499-501: "However, a correlation of TEP to wind speed was not found. It may be that since wind speed data represented an average value of 24 hours, short but pronounced changes in the wind speed were not visible in the average wind speed value."

10. II. Thus the increase in wind speed is directly related to the increase in sea salt concentrations in aerosols. Perhaps the same process applies to TEP, hence its higher concentrations in in the ambient atmosphere than from the plunging waterfall tank were noted. Lewandowska and Falkowska (2013) determined that the amount of sea salt transferred into the near-water layer of the atmosphere increases exponentially already with wind speed over 3 m·s$^{-1}$. The limit value over sea was the same as suggested in literature (Nair et al, 2005; Meira et al, 2007).

Regarding the tank samples, the waterfall is simulating the bubble bursting (and the wind speed). We cannot translate the waterfall intensity and the resulting sea spray formation to a certain wind speed for a comparison with the ambient conditions. However, sodium concentrations in the tank were higher than in the atmospheric concentration, which shows that there was a strong sea spray generation. At the same time, TEP concentrations were significantly lower in the tank (compared to ambient conditions), suggesting that bubble bursting is not the major driver for TEP on the aerosol particles, supporting the idea of a secondary formation in the atmosphere. We underlined this in the manuscript more strongly by adding the following text:

Line 450-455: "It should be noted that the lower enrichment in the tank resulted from the lower TEP number concentrations in the generated aerosol particles, as the particulate sodium concentrations in the tank aerosol were even higher than in the ambient particles (Tab. S3). This suggests that, although an artificial tank study cannot represent the ambient environment, the generation of sea spray aerosol was in progress; however, TEP transfer seemed to be not pronounced. "

To underline the effect of wind speed on the sea spray aerosol production and potential TEP transfer, we added the reference of Lewandowska and Falkowska (2013) and others in the introduction as follows:

Line 108-111:  "Due to wind and breaking waves, sea spray aerosol particles are formed (de Leeuw et al., 2011;Lewandowska and Falkowska, 2013;Liss and Johnson, 2014) that could be a transfer mechanism for TEP from the ocean to the atmosphere."

-1

11. Lines 538-539- Remove the sentence "The pH  in the cloud water analysed here was between 6.3 and 6.6." and in Line 541 I propose to change the sentence like this:- "At cloud water pH-values was between 6.3 and 6.6, and marine gels could split into smaller units (Chin et al., 1998), that are below the minimum detectable particle size of 4.5 µm."

We agree and rephrased these sentences. However, we wanted to differentiate between marine gels (mentioned by Chin et al. 1998) and the TEP analysed here. It now reads:

Line 578-581: "The measured cloud water pH-value of the samples analysed here was between 6.3 and 6.6, at which marine gels could split into smaller units (Chin et al., 1998). Hence, a part of the cloud water TEP might be below the minimum detectable particle size of 4.5 µm due to the slightly acidic conditions."

12. Line 541- In my opinion it should be: "... the different factors such as pH, ion density ..." or  "..the variables pH, ion density…"

We agree and changed to: Line 577:  "…might be affected by the different factors such as pH, ion density, temperature and pressure in the atmosphere."

13. Line 547- I propose like this: "…fully explain the role of each of these effects but our investigations…".

We agree and changed to: Line 584-585: "from our data we cannot fully explain the role of each of these effects and such investigations warrant further studies".

14. In Chapter 3.3.2.2 (Biotic formation)- Authors indicated that TEP can be directly released as particulates from aquatic organisms involving phytoplankton and bacteria (Lines 551-552). However, cyanobacteria and microalgae, which are also present in the air, have been omitted. An entire chapter is devoted to bacteria. While there are no reports on cyanobacteria and microalgae in cloud water, there have been many publications on their presence in aerosols recently (e.g. Sharma et al., 2007; Genitsaris et al., 2011; Després et al., 2012; Sahu and Tangutur, 2014; Lewandowska et al., 2017; WiÅ›niewska et al., 2019 and much more). It is also worth mentioning them in this publication, even if it were only a few sentences. Especially that in the introduction Authors mentioned that high TEP concentrations can be associated with phytoplankton (mainly diatom) blooms. Perhaps the considerations in Chapter 3.4 (Lines 626- 637) regarding bacteria could also apply to cyanobacteria and microalgae and their metabolic degradation products that occur in aerosols? Such a reflection for the future.

We agree that cyanobacteria and microalgae and their metabolic degradation products that occur in aerosols might contribute to atmospheric TEP processing. We thank the reviewer for his constructive input and took up this interesting thought, adding the following text:

Line 622-626: "Besides, although not measured here, microalgae and cyanobacteria, that are relevant for direct TEP formation in seawater, have been reported to occur in the atmosphere (e.g. Lewandowska et al., 2017;Sharma et al., 2007;Wiśniewska et al., 2019;Wiśniewska et al., 2022). It is worth studying, if these species and their metabolic degradation products contribute to atmospheric TEP processing."

Line 709-713: Finally, while dust might be a dominant INP source in the here investigated tropical Atlantic region close to the Saharan desert, in other remote oceanic locations, marine gel particles, their in-cloud formation and connection to bacteria **and phytoplankton** in the atmosphere could be highly relevant for a better understanding of marine cloud properties.

**Conclusions** are fine to me.

**Caption of Figures** are comprehensive and, in my opinion, correct. The same for **tables**.

**Figures:**

15. **Fig. 1**- Enlarge so that the scale in the drawings was visible

The resolution of the Figures appear partly poor due to the requested portrait mode. For publication, they will be provided in the best possible format and resolution in a separate document to ensure readability.

16. **Fig. 2**- TEP concentrations were below the limit of detection (LOD) only on 26th of September 2017 - as shown in the picture. And also: "…the three cloud water samples (blue-red squares)"- it looks black - red rather than blue – red squares.

Thanks for noting these issues. The dates were corrected and the colour description was corrected.

17. **Fig. 5-** Please give superscripts on the oy axis [mL$^{-1}$]

Done

18. **Fig. 6-** Remove the comma in front of the [%] in the oy axis. The % values for the range 5.5 to 7 μm should be given above the bars as for the other bars

Done

19. **Fig 7-** Increase the descriptions for both axes in all figures

We increased the axis descriptions. As mentioned above, the resolution of the Figures appear partly poor due to the requested portrait mode. For publication, they will be provided in the best possible format and resolution in a separate document to ensure readability.

20. **Literature**- Align to margins and validate against journal guidelines

Done

**Citation**: https://doi.org/10.5194/acp-2021-845-RC1

Cited Literature:

- Chin, W. C., Orellana, M. V., and Verdugo, P.: Spontaneous assembly of marine dissolved organic matter into polymer gels, Nature, 391, 568-572, 10.1038/35345, 1998.
- de Leeuw, G., Andreas, E. L., Anguelova, M. D., Fairall, C. W., Lewis, E. R., O'Dowd, C., Schulz, M., and Schwartz, S. E.: Production Flux of Sea Spray Aerosol, Reviews of Geophysics, 49, 10.1029/2010rg000349, 2011.

- Engel, A.: Determination of Marine Gel Particles in: Practical guidelines for the analysis of seawater, edited by: [u.a.], O. W. B. R., CRC Press, 2009.
- Lewandowska, A. U., and Falkowska, L. M.: Sea salt in aerosols over the southern Baltic. Part 1. The generation and transportation of marine particles, Oceanologia, 55, 279-298, 10.5697/oc.55-2.279, 2013.
- Lewandowska, A. U., Sliwinska-Wilczewska, S., and Wozniczka, D.: Identification of cyanobacteria and microalgae in aerosols of various sizes in the air over the Southern Baltic Sea, Marine Pollution Bulletin, 125, 30-38, 10.1016/j.marpolbul.2017.07.064, 2017.
- Liss, P. S., and Johnson, M. T.: Ocean-Atmosphere Interactions of Gases and Particles, Springer, 2014.
- Logan, B. E., Passow, U., Alldredge, A. L., Grossartt, H.-P., and Simont, M.: Rapid formation and sedimentation of large aggregates is predictable from coagulation rates (half-lives) of transparent exopolymer particles (TEP), Deep Sea Research Part II: Topical Studies in Oceanography, 42, 203-214, https://doi.org/10.1016/0967-0645(95)00012-F, 1995.
- Mari, X., Passow, U., Migon, C., Burd, A. B., and Legendre, L.: Transparent exopolymer particles: Effects on carbon cycling in the ocean, Progress in Oceanography, 151, 13-37, 10.1016/j.pocean.2016.11.002, 2017.
- Sharma, N. K., Rai, A. K., Singh, S., and Brown, R. M.: Airborne algae: Their present status and relevance, Journal of Phycology, 43, 615-627, 10.1111/j.1529-8817.2007.00373.x, 2007.
- Wiśniewska, K., Lewandowska, A. U., and Śliwińska-Wilczewska, S.: The importance of cyanobacteria and microalgae present in aerosols to human health and the environment - Review study, Environment International, 131, 10.1016/j.envint.2019.104964, 2019.
- Wiśniewska, K. A., Śliwińska-Wilczewska, S., and Lewandowska, A. U.: Airborne microalgal and cyanobacterial diversity and composition during rain events in the southern Baltic Sea region, Scientific Reports, 12, 2029, 10.1038/s41598-022-06107-9, 2022.
- Wurl, O., and Holmes, M.: The gelatinous nature of the sea-surface microlayer, Marine Chemistry, 110, 89-97, 10.1016/j.marchem.2008.02.009, 2008.
- Wurl, O., Miller, L., and Vagle, S.: Production and fate of transparent exopolymer particles in the ocean, J. Geophys. Res.-Oceans, 116, 10.1029/2011jc007342, 2011.

---

## Author Comment (AC2)

Referee 2:

Comment on "High number concentrations of transparent exopolymer particles (TEP) in ambient aerosol particles and cloud water – A case study at the tropical Atlantic Ocean" by Manuela van Pinxteren et al.

1. The manuscripts deals with an interesting and quite new topic and presents some potentially important hypothesis and scientific questions (Are "secondary" TEPs more important than "primary" ones in the marine atmosphere? Is the marine aerosol TEP population connected in any way to the INP population?). The paper is well written and sufficiently clear. I recommend publications once the following (minor) issues are clarified.

   We thank the Reviewer for the evaluation and the constructive comments. We like the term "secondary" formation in relation to TEP that was introduced by the reviewer and included it in the manuscript (abstract and conclusion).

   Replies to the specific Referee's comments are provided below in red and new parts included in the manuscript are marked in **green**. Line numbers refer to the revised (clean) version.

2. L40. The TEP concentrations reported do not represent supermicron aerosols. They correspond to particles larger than 4.5 microns (as correctly stated above), which represents a subset of the supermicron TEP population. Please modify for major accuracy.

   We agree with the reviewer that not the entire population of supermicron particles was covered in the analysis and corrected it accordingly. In the abstract, we included this aspect:

   Line 37-40:"Here, we report number concentrations of TEP with a diameter> 4.5 µm, **hence covering a part of the supermicron particle range**, in ambient aerosol and cloud water samples from the tropical Atlantic Ocean as well as in generated aerosol particles using a plunging waterfall tank that was filled with the ambient seawater."

3. L42-43. I would suggest to add that the conversion was based on the observed cloud LWC.

   This was included and now reads:

Line 42-45: "Cloud water TEP concentrations were between $4 \times 10^6$ and $9 \times 10^6$ #TEP L$^{-1}$ and, according to the measured cloud liquid water content, corresponding to equivalent air concentrations of $2 - 4 \times 10^3$ #TEP m$^{-3}$."

4.  L43-46. The TEP concentration in the tank headspace has no atmospheric relevance. It results from the experimental parameters chosen to operate the tank and can be modified just by varying them (e.g. headspace flush flow, intensity of the plunging jets, and so on...). The only general and valuable information that can be extracted from sea-spray tanks regards the (size-dependent) relative chemical composition of the produced sea-spray particles. I am not against presenting the obtained TEP concentration in the tank, but it should not be reported in the abstract. More comments on this issue follow below.

    We agree and deleted the tank concentration from the abstract and only included the enrichment factors. In addition, we included more information on the tank aerosol as reported in detail below. Please see our responses to comments No. 6 and 10.

5.  L81. Instead of "contain" maybe "have" would be more correct?

    We agree and changed the term accordingly.

6.  L303-321. Please clarify where the seawater TEP concentration used to derive the EFs comes from. Reading further on, one understands that it derives from Engel et al. (2020). For major clarity, this information should be added here.

    We agree that the reference of the origin of the seawater TEP concentration is an important point. The seawater TEP data were achieved from Engel et al. (2020), which was stated in Table 1. We added the following text (that was originally put in the supporting information):

    Line 420-423: "However, the TEP number concentrations in the ocean surface water were obtained from an additional measurement campaign, taking place in the biologically productive Mauritanian Upwelling region in the year 2012, hence at another time and season (Tab. 1)."

7.  L330-333. Please, provide a more quantitative comparison between the TEP concentration in aerosol and cloud water. If the authors think that this is not possible, they should re-consider the following sentence: "suggesting that the majority of the TEP particles are activated to cloud droplets when a cloud forms". Judging by the plot in Figure 2 and comparing the cloud cases with the corresponding aerosol samples (which is

a very raw approach), I would say that less than 30% of TEPs are activated into a cloud. This is in contrast with the above statement. If the authors have data to support their statement, I would invite them to discuss them quantitatively. By the way, the highlighted statement seems to be contradicted by the authors themselves in Lines 543-545.#

We thank the reviewer for this comment. This was clearly a mistake and the term "the majority of the TEP are activated to cloud droplets" was not correct. Unfortunately, the cloud water sampling cases for TEP measurements were limited. However, as the reviewer suggested, we compared the #TEP concentrations in cloud water to the #TEP concentrations in the ambient aerosol particles when sampling times coincided, i.e. on 20.09., 28.09., and 04.10. 2017. For these dates, the cloud water #TEP concentrations made up between 10 and 34% (average: 20%) of the #TEP ambient aerosol concentrations. In addition, we related the average cloud water #TEP concentrations to the average #TEP concentrations in the ambient aerosol, which showed that the cloud water #TEP concentrations made up 24% of the #TEP ambient aerosol concentrations. Regarding these numbers, we conclude that the #TEP concentrations in cloud water were **about 20%** of the #TEP concentrations in the ambient aerosol (with a good agreement regarding the matching dates and the average values). This finding is now in line with the statement that the reviewer referred to in Line 543-545 *"the lower concentrations in cloud water ($2 - 4x10^3$ #TEP $m^{-3}$) compared to ambient aerosol particles ($7x10^2 - 3x10^4$ #TEP $m^{-3}$)."*

We revised the manuscript as follows:

Line 336-338: "Comparing the #TEP concentrations in cloud water to the ones in the ambient aerosol particles suggested that about 20% of the ambient TEP particles are activated to cloud droplets when a cloud forms."

8. L336-341. The concentration of TEP in the sea-spray aerosol generated by the tank is not an atmospheric relevant information; it depends only on the chamber design and settings. Therefore, there is no reason to compare the tank TEP concentration with that of atmospheric samples, either by performing a statistical test or not. The fact that the chamber produced lower concentrations of TEP with respect to what observed in ambient samples is irrelevant; the only informative data available from the tank are the EF data (because they are based on the relative chemical composition i.e., TEP/Na+), which the authors use correctly to infer about TEP sources later on in the text.

We agree that tank studies cannot represent the ambient environment and on the caveats regarding the chamber design etc. However, tank studies are helpful for regarding the ocean-atmosphere transfer via (only) bubble bursting by eliminating aerosol processing and additional aerosol sources. As the reviewer suggested, we rather focus on the comparison of the enrichment factors of the tank experiments with the ambient studies. In addition, we added the information on the tank data; please see our detailed answers to comment No. 10.

9. L347. "besides for the…". Please double check English.

We rephrased this sentence to "In addition to…"

10. L429-433. [Par. 3.2]. Which seawater TEP concentration was used to calculate the EF for the sea-spray tank samples? Where TEP directly measured in the same seawater used for the bubbling experiments? Or did the authors use the average ETNA values from Engel et al. (2020)? Please provide this information here and in the Experimental Section. This may be quite a critical point. Considering how variable biochemical parameters may be in seawater, assuming that the average TEP concentration measured by Engel et al. (2020) is representative of the water samples used for the bubbling experiments is quite risky. The comparison of TEP EFs between atmospheric and tank samples is the only solid base that supports the (very interesting) hypothesis of important "secondary" TEP formation processes in the atmosphere. If the EF calculated for the tank samples are biased by assuming a seawater TEP concentration that does not truly represent the real situation in the tank, this base appears much less solid. In this case, I must invite the authors to add some caveat in the text, making it clear to the readers that the hypothesis, although very interesting, needs to be further demonstrated as doubts still persist at this stage given the uncertainties inherent to the EF calculations for tank samples.

The reviewer raised an important point here. As stated above, the seawater TEP data were obtained from Engel et al. 2020. We added an error discussion with regards to the oceanic TEP measurements and pointed out that the here reported $EF_{atm}$ represent a lower limit. We want to underline that even though absolute numbers can vary (due to the potential biases resulting from the seawater data), the strong differences between the $EF_{aer. ambient}$ and $EF_{aer. tank}$ (Tab. S3) are evident, as they result from the same type of seawater.
In addition, in the revised version, we compared the total TEP number concentrations used in this study to previous measurements performed at the Cape Verde islands (south of São Vicente at 16°44.4'N, 25°09.4'W) and found a difference of a factor of 2. However, from the study of Engel et al. (2015) solely the total number concentrations are available. In the present work we exclusively used the number concentrations in the size range between 4 and 10 µm (to cover the same range as used for the atmospheric TEP size ranges) and therefore, we cannot use the values from Engel et al. 2015 for calculating the enrichment factor and kept the original values from the ETNA (Engel et al. 2020). We included this comparison in the revised version as shown below.

The following information was originally placed in the Supporting information, but as the reviewer rightly stated, they are important for classifying the results and were consequently added to the revised manuscript as follows:

Line 423-434: "Compared to other oceanic regions, the TEP values from the Mauritanian Upwelling region were at the higher end (Engel et al., 2020). The region around the CVAO is rather oligotrophic and Chlorophyll-a values during the MarParCloud campaign were relatively low with 0.1 up to 0.6 µg L$^{-1}$ (van Pinxteren et al., 2020). As TEP production is often connected to phytoplankton activity, the TEP concentration at the CVAO might be lower compared to more productive regions (Robinson et al., 2019a). A previous study showed the total TEP number concentrations (covering TEP sizes between 1 and 200 µm) at the Cape Verde islands (south of São Vicente at 16°44.4'N, 25°09.4'W)

were by a factor of 2 lower that the data reported here, in detail $0.9 \times 10^7$ L$^{-1}$ (Engel et al., 2015) vs. $2 \times 10^7$ L$^{-1}$ (Tab. S5). Lower TEP concentrations would result in higher EF$_{atm.}$ (Equation 1) regarding the ambient as well as the tank measurements as the same type of seawater was used for the calculations. Hence, the here reported EF$_{atm.}$ represent lower limits."

In addition, we like to point out that the "sea spray generation" in the tank was successful, as the sodium concentrations in the tank aerosol particles were even higher than the ambient sodium concentrations, but TEP concentrations were comparably low in the tank aerosol. This is an important information that we did not make clear in the original manuscript. We added the following text:

Line 450-455: "It should be noted that the lower enrichment in the tank resulted from the lower TEP number concentrations in the generated aerosol particles, as the particulate sodium concentrations in the tank aerosol were even higher than in the ambient particles (Tab. S3). This suggests that, although an artificial tank study cannot represent the ambient environment, the generation of sea spray aerosol was in progress; however, TEP transfer seemed to be not pronounced."

Furthermore, we like to underline that the higher *EF*$_{aer.\ ambient}$ compared to EF tank is not the only solid base that supports the hypothesis of the "secondary" TEP formation processes. The finding that the *EF*$_{aer.\ ambient}$ are really high for particles with a diameter> 4.5 μm points to additional processes (besides bubble bursting), as such high enrichment for this aerosol particles size is uncommon and larger than previously reported in the literature, indicating additional (secondary) processes. This is summarized in the discussion Lines 458-471.

We think that with the applied changes we have addressed this issue more clearly and showed the limitations of the study. In the conclusion, we underlined the strong need for additional studies regarding this topic.

11. L479-484. "the high abundance and enrichment of #TEP in the ambient aerosol particles compared to literature data (Kuznetsova et al., 2005) and compared to the concentration and enrichment of the #TEP from the plunging waterfall tank performed here, suggests that additional TEP sources in the ambient atmosphere exist from which TEPs are added to their primary transfer by bubble bursting from the oceans". I stress again that the difference in concentration of TEP between atmospheric and tank samples is not supporting the existence of secondary TEP formation processes in the atmosphere. It is only the result of the tank properties. Only the difference in the TEP enrichment with respect to Na between tank and atmosphere supports this. Please, remove the references to "abundance" and "concentration" in the above sentence.

We thank the reviewer for this comment. To account for this, in the revised version we have prioritized our focus on the enrichment factors rather than on the (tank) concentrations (see our detailed answers to question No. 10). The major reasons are the

high enrichment of ambient TEP compared to tank TEP (while there was strong evidence that sea spray generation was in progress) as well as the generally high TEP ambient enrichment compared to literature studies. We underlined that additional studies are strongly needed.

As suggested by the reviewer we deleted the reference from this sentence.

12. Par3.3.2. This paper focuses on atmospheric particles larger than 4.5 µm, which are characterized by fairly short atmospheric residence times. It may be worth discussing if the atmospheric lifetime of these particles is consistent with the timescale of TEP (biotic and abiotic) formation reactions. Data should be available at least for the seawater compartment if not for the atmosphere.

We picked up this interesting thought and in the revised version, we included references that investigated abiotic and biotic transformation of TEP and similar substances (EPS). We compared them with the lifetime of supermicron aerosol particles (where the here investigated TEP are part of). We found that the transformation rates are within the lifetime of aerosol particles and therefore concluded that these processes might be relevant for TEP formation in the atmosphere, although care must be taken when transferring ocean processes to atmospheric processes and further studies are required on this topic.

We included the following parts for the abiotic and biotic processing:

Line 535-551: "In the ocean, the dissolved polysaccharides are known TEP precursors (Passow, 2002) and Wurl et al. (2011) determined abiotic TEP formation rates from dissolved polysaccharide concentration in various oceans. The rates were on average 7.9 ± 5.0 µmol C $L^{-1}$ $d^{-1}$ and therefore significant, considering that the average TEP concentration was 18.1 ± 15.9 µmol C $L^{-1}$ and the average dissolved polysaccharide concentration was 12.2 ± 3.8 µmol C $L^{-1}$ in the surface seawater (Wurl et al., 2011). Robinson et al. (2019b) showed that rising bubbles can lead to an enhanced TEP formation already after some minutes. The lifetime of supermicron aerosol particles, to which the TEP particles studied here belong, are between hours and days, for example, Madry et al. (2011) calculated an average lifetime of supermicron sea salt particles of 50 hours. Hence, abiotic TEP formation processes lie within the lifetime of supermicron aerosol particles and we suggest that spontaneous TEP formation from the (high) abundant dissolved polysaccharides likely contributed to the high TEP concentrations observed in the ambient atmosphere in the present study. However, it needs to be considered that the abiotic TEP formation processes as described by Wurl et al. (2011) and Robinson et al. (2019b) were relevant for the oceanic environment and might not directly translated to atmospheric processes. Further studies are required on this topic."

and

Line 615-619: "Regarding time scales of biotic processing, Matulova et al. (2014) showed that the Bacillus sp. 3B6, isolated from cloud water, was able to bio-transform saccharides that are present in the atmosphere. The saccharides formed exopolymer substances (EPS), of which TEP are a subgroup. The formation of EPS was revealed after 48 h of incubation and therefore within the lifetime of supermicron aerosol particles (Madry et al., 2011)."

**Citation**: https://doi.org/10.5194/acp-2021-845-RC2

Cited Literature:

Engel, A., Borchard, C., Loginova, A., Meyer, J., Hauss, H., and Kiko, R.: Effects of varied nitrate and phosphate supply on polysaccharidic and proteinaceous gel particle production during tropical phytoplankton bloom experiments, Biogeosciences, 12, 5647-5665, 10.5194/bg-12-5647-2015, 2015.

Engel, A., Endres, S., Galgani, L., and Schartau, M.: Marvelous Marine Microgels: On the Distribution and Impact of Gel-Like Particles in the Oceanic Water-Column, Frontiers in Marine Science, 7, 10.3389/fmars.2020.00405, 2020.

Madry, W. L., Toon, O. B., and O'Dowd, C. D.: Modeled optical thickness of sea-salt aerosol, Journal of Geophysical Research-Atmospheres, 116, 10.1029/2010jd014691, 2011.

Matulova, M., Husarova, S., Capek, P., Sancelme, M., and Delort, A. M.: Biotransformation of Various Saccharides and Production of Exopolymeric Substances by Cloud-Borne Bacillus sp 3B6, Environmental Science & Technology, 48, 14238-14247, 10.1021/es501350s, 2014.

Passow, U.: Transparent exopolymer particles (TEP) in aquatic environments, Progress in Oceanography, 55, 287-333, 10.1016/s0079-6611(02)00138-6, 2002.

Robinson, T. B., Stolle, C., and Wurl, O.: Depth is relative: the importance of depth for transparent exopolymer particles in the near-surface environment, Ocean Science, 15, 1653-1666, 10.5194/os-15-1653-2019, 2019a.

Robinson, T. B., Wurl, O., Bahlmann, E., Juergens, K., and Stolle, C.: Rising bubbles enhance the gelatinous nature of the air-sea interface, Limnology and Oceanography, 64, 2358-2372, 10.1002/lno.11188, 2019b.

van Pinxteren, M., Fomba, K. W., Triesch, N., Stolle, C., Wurl, O., Bahlmann, E., Gong, X. D., Voigtlander, J., Wex, H., Robinson, T. B., Barthel, S., Zeppenfeld, S., Hoffmann, E. H., Roveretto, M., Li, C. L., Grosselin, B., Daele, V., Senf, F., van Pinxteren, D., Manzi, M., Zabalegui, N., Frka, S., Gasparovic, B., Pereira, R., Li, T., Wen, L., Li, J. R., Zhu, C., Chen, H., Chen, J. M., Fiedler, B., Von Tumpling, W., Read, K. A., Punjabi, S., Lewis, A. C., Hopkins, J. R., Carpenter, L. J., Peeken, I., Rixen, T., Schulz-Bull, D., Monge, M. E., Mellouki, A., George, C., Stratmann, F., and Herrmann, H.: Marine organic matter in the remote environment of the Cape Verde islands - an introduction and overview to the MarParCloud campaign, Atmospheric Chemistry and Physics, 20, 6921-6951, 10.5194/acp-20-6921-2020, 2020.

Wurl, O., Miller, L., and Vagle, S.: Production and fate of transparent exopolymer particles in the ocean, J. Geophys. Res.-Oceans, 116, 10.1029/2011jc007342, 2011.